# Economic costs of terminal care for selected non-communicable diseases from a healthcare perspective: a review of mortality records from a tertiary hospital in Nigeria

Adesola Oluwafunmilola Olumide ![ORCID],[1] Amir Shmueli,[2] Olayemi O Omotade,[1] Emmanuel S Adebayo,[3] Temitope O Alonge,[4] Gabriel O Ogun[5]

For numbered affiliations see end of article.

**Correspondence to**
Dr Adesola Oluwafunmilola Olumide; daisyolu@yahoo.co.uk

## ABSTRACT

**Introduction** WHO revealed that morbidity and mortality from non-communicable diseases (NCDs) are on the increase and NCDs accounted for approximately 29% of all deaths in Nigeria in 2016. This study was conducted to estimate the economic cost of selected NCDs—lung cancer, liver cancer and liver cirrhosis. These diseases are known to be associated with key modifiable health risk behaviours (smoking and alcohol use), which are prevalent in Nigeria and often commence during the adolescent years.

**Methods** Data were obtained between 2016 and 2017, from mortality records of patients managed for the selected diseases in the University College Hospital, a major referral centre in Nigeria. Information on costs of treatment, clinic visits, admission and transportation was obtained. Average costs of terminal in-patient care and transportation costs (in 2020 prices) were computed per patient. Costs were converted to the US dollar equivalent using the current official rate of US$1: ₦360.50.

**Results** Twenty-two (out of 90 cases recorded) could be retrieved and all the patients had been diagnosed in the terminal stages of the disease. The average direct costs were ₦510 152.62 (US$1415.13) for an average of 49.2 days of terminal care for lung cancer; ₦308 950.27 (US$857.00) and ₦238 121.83 (US$660.53) for an average of 16.6 and 21.7 days of terminal care for patients managed for liver cancer and liver cirrhosis, respectively.

**Conclusion** The economic costs of each of the diseases were very high. Findings emphasise the need for aggressive efforts to promote primary prevention, improve early diagnosis and provide affordable treatment in view of the fact that the monthly minimum wage is less than US$85.00 and treatment costs are borne out-of-pocket by the generality of the population in Nigeria.

## INTRODUCTION

Globally, there has been an increase in the prevalence of non-communicable diseases (NCDs).[1] These diseases were previously viewed as diseases of the affluent with a high prevalence in developed countries.[2] This is

### Strengths and limitations of this study

► This is a novel study providing direct costs of terminal care for selected non-communicable diseases (NCDs) in Nigeria from a healthcare perspective.
► The data collection methods used were robust; lending credence to the strength of the findings.
► Costs are based on data retrieved from a Federal government tertiary hospital in Nigeria which are lower than equivalent costs obtainable in private facilities.
► Costs could only be computed for the 22 records retrieved (representing 24.4% of all related mortalities within the time frame studied).
► In spite of the small sample, the findings are reasonable estimates of direct costs of terminal care for the selected NCDs in the study setting and are an important first step to further enquiry.

no longer the case however and the larger burden of these diseases is now borne by low-income and middle-income countries.[1] While still grappling with the burden from communicable diseases like malaria and pneumonia, low-income and middle-income countries are experiencing an increase in the occurrence of chronic NCDs.[1 3] Thus, these countries are currently experiencing a double burden of diseases from communicable and NCDs.[1 3] Research has proven that many of the chronic conditions that individuals suffer in adulthood are linked to unhealthy habits, for example, substance including tobacco and alcohol use, unhealthy eating habits and unsafe sexual intercourse that they adopted during their younger years.[4–6] Common established outcomes of these health risk behaviours include the occurrence of various cancers and other non-communicable conditions such as cardiovascular diseases (CVDs)

and diabetes later in life.[1] These diseases result in considerable morbidity, mortality and economic losses[7] emphasising the need for continued research to inform policy and practice geared towards their control.

In Nigeria, the disease burden from communicable diseases is still very high, however, available data indicate that the burden from NCDs is significant. WHO estimated that morbidity and mortality from NCDs accounted for approximately 29% of all deaths in the country in 2016.[8] According to 2020 data from the Global cancer observatory, there were about 124 815 new cases of cancer in Nigeria. Liver and lung cancer were among the top 20 causes of cancer in Nigeria with incidence rates of 4.4% and 1.4%, respectively (based on weighted/simple average of the most recent local rates applied to 2020 population).[9] Furthermore, of the estimated 78 899 deaths from cancer, 5180 and 1789 were from liver and lung cancer respectively. Adekanle *et al* in their review of records of patients admitted with liver diseases into a tertiary hospital in Ile–Ife, Nigeria, from 2013 to 2017 found that hepatocellular carcinoma and liver cirrhosis accounted for 52.8% and 27.2% of all cases of liver disease admitted during the review period.[10]

Cost-of-illness studies determine the burden of diseases to society in monetary terms. In addition, they provide an estimate of the amount that could potentially be saved if a disease were to be eradicated.[11–13] Typically, cost-of-illness studies do not evaluate interventions that address the disease under study. Thus they do not provide information on the benefits of implementing specific interventions or which of a range of interventions would produce maximal impact.[13] Cost-of-illness studies as the name implies, document a range of illness costs which are usually categorised as direct, indirect or intangible costs. Direct costs measure, 'the opportunity cost of resources used for treating a particular illness' or, 'costs for which payments are actually made'. Examples of direct costs could be direct medical costs such as costs of treatment, drugs, in-patient admission or direct non-medical costs such as transportation costs, costs of special meals, relocation costs etc.[13 14] Indirect costs refer to, 'the value of resources lost due to a particular illness', and intangible costs include costs of pain and suffering.[13 14]

There has been an increase in the numbers of studies documenting the cost of various NCDs, however, the overwhelming majority of these studies have been conducted in developed countries or in low-income and middle-income countries in Asia.[15–21] Data from existing studies confirm the immense economic burden from NCDs. Muka *et al* in their review of the global impact of NCDs on healthcare found that globally, cancer and CVDs had the highest reported mean annual total direct costs per patient of approximately US$197 772 and US$81 096, respectively.[22] A systematic review of cost of diabetes mellitus in Africa by Mutyambizi *et al* reported national direct costs of diabetes ranging from 3.5 to 4.5 billion international dollars per annum.[23] There remains a paucity of these studies in sub-Saharan African and in Nigeria, resulting in a reliance on cost estimates from developed countries or low-income and middle-income countries outside of sub-Saharan Africa for drawing inferences for the economic burden of diseases in our region. The current study was conducted to determine the cost of selected chronic illnesses, namely, lung cancer, liver cancer and chronic liver disease in Ibadan Nigeria. These illnesses were selected because they are recognised long-term consequences of some modifiable health risk behaviours associated with a significant proportion of NCDs, namely smoking and alcohol consumption, they result in significant morbidity and mortality and there are proven measures for primary prevention of the health risk behaviours they are associated with. Data were obtained as part of a larger study on the predictors and economic costs of selected health risk behaviours among adolescents in Oyo state, Nigeria.

## METHODS

This prevalence-based study adopted a healthcare system perspective and thus focused on the medical costs of the illnesses. The estimation of the cost of care for the long-term illnesses in this study was guided by the cost-of-illness approach described by Rice[24] and the WHO economic framework of cost estimation described in the WHO guide to identifying the economic consequences of disease and injury[15 25]

Data on the illness costs were collected between September 2016 and January 2017 from the University College Hospital (UCH), Ibadan. Established in 1952, it is the premier teaching hospital in Nigeria and is located in Ibadan, South west Nigeria. The hospital also houses the first cancer registry in Nigeria—the Ibadan Cancer Registry which was established in 1960 within the Pathology department of the University of Ibadan.[26] The UCH is a training centre for doctors, nurses and various other categories of health staff at undergraduate and postgraduate levels and also provides tertiary level health services. It is a major receiving centre for tertiary level management of patients with various communicable and NCDs who are referred from various parts of the country.

We obtained data retrospectively from a review of case records of patients who had been diagnosed and received treatment for the illnesses investigated. The Case note numbers of patients who died from the diseases of interest—lung cancer and liver disease (liver cancer and liver cirrhosis) in 2014 and 2015 were obtained from the Medical records department. There were a total of 12 reported deaths from lung cancer, of these, 5 case notes could be retrieved. In all, 78 patients were reported to have died from liver diseases (liver cirrhosis, liver cancer and chronic liver disease) in 2014 and 2015. Of these, nine case notes of patients with liver cancer and eight for patients with liver cirrhosis could be retrieved. Nine case-files of patients with chronic liver disease were retrieved but information from these cases notes were not used because there was no definitive diagnosis in the case note

(this could have occurred because payment was yet to be made for the investigations, hence they were not carried out). Ladep and Taylor-Robinson had noted that some challenges to a definitive diagnosis of liver cirrhosis in Nigeria were reluctance to give consent for liver biopsy and inadequate numbers of trained doctors to conduct the procedure.[27] Relevant information—age, sex, home address, date of diagnosis, medical treatment, investigations, number of clinic visits, admission history (number of admissions and duration on admission) were extracted manually from the patient case notes onto a datasheet.

Direct medical (healthcare) costs obtained included the following: cost of hospital admission, clinic visits, investigations, medical procedures, medications and consumables. The time from presentation with symptoms to demise for the patients whose case records were obtained ranged from zero to about four and a half months indicating that most patients were admitted at the point when the illnesses were at the terminal stages. Costs were obtained from the relevant departments in the hospital. In addition, costs of investigations and drugs were also obtained from a registered private laboratory and a pharmacy within Ibadan. This was because anecdotal reports indicated that some patients procure their drugs or get some of their investigations done outside the hospital. The costs of drugs and investigations from these private sources were compared with hospital costs, summed and the average for each item was documented. For each illness investigated (lung cancer, liver cancer and liver cirrhosis), the total medical costs for each patient was summed and the average costs incurred from the time of diagnosis or presentation with symptoms to demise was estimated for each disease based on information obtained from the patient's case note.

The direct non-medical cost of illness was limited to transportation costs only. Transportation costs were computed for each patient using the home address provided in the case note. A list of all the addresses was compiled and information on transportation costs from these neighbourhoods to the hospital was obtained from informal interviews with commercial transport drivers in the relevant motor parks. The cost of regular public transportation and the cost of car hire services were obtained. For public transportation, the costs of a return trip for two people was calculated as it was assumed that at least one person would accompany the patient to hospital at every visit because information obtained from the case notes indicated that the patients presented quite late in the course of the disease. For car hire, the cost of hiring the vehicle for a return trip was computed. The average costs (public transportation and car hire) were computed for each patient and the average transportation costs per patient calculated for each disease (see online supplemental appendix 1 for table showing average costs per item for each of the three diseases)

For the purposes of this paper, all costs are reported using 2020 as the base year. The 2017 prices were converted to their 2020 values using the average annual inflation rate reported by the Central Bank of Nigeria (CBN) between 2017 and May 2020 which was 13.23%.[28] The costs were then converted to their dollar equivalent using the official CBN June 2020 exchange rate of ₦360.50: US$1.00[29] to aid comparison with findings from other studies.

Costs were computed using the formulae below:
1. Total direct cost of care (DC) for all patients for duration of treatment in days
   =direct medical (DM) + direct non-medical costs (DN-M)
2. Average direct cost for all the patients for the duration of treatment in days (ADC)
   =DC/number of patients (n).

## Patient and public involvement statement
Patients or the public were not involved in the design, or conduct, or reporting, or dissemination plans of our research.

## RESULTS
The mean age (and SD) of the five patients (three male and two female) with a diagnosis of lung cancer was, 59.8 (±10.6) years. The patients (eight males and one female) with liver cancer had an average age of 52.6 years (SD=7.9). There were eight male patients with liver cirrhosis and they had a mean age of 44.3 years (SD=13.1).

### Medical costs of lung cancer
Records retrieved revealed that on the average, the patients were managed for 49.2 days (from presentation with symptoms to demise) and the average number of days in total spent on admission in hospital was 26.7 days (range: 1–90 days). Patients made an average of 1.6 trips to the hospital (range=1–3 trips). The average direct medical costs of managing one patient for 49.2 days in 2020 prices was ₦484 445. 46 (US$1343.82) and transportation cost was ₦25 707.16 (US$71.31). Average direct (medical and non-medical) cost of managing one patient was about ₦510 152.62 (US$1415.13); (table 1).

### Medical cost of liver cancer
The average number of days in total spent in hospital across all of the patients' admissions was 16.6 days (1– 77). Details of the average direct costs for managing a patient with liver cancer (from presentation with symptoms to demise) are presented in table 1. Average direct medical cost was ₦300.764.15 (US$834.30) per patient and average transportation cost was ₦8186.12 (US$22.71), giving average direct medical and non-medical cost of ₦308 950.27 (US$857.00).

### Medical costs of liver cirrhosis
Details of the average direct costs for managing a patient with liver cirrhosis in 2020 prices are presented in table 1. The eight patients for whom information was obtained were managed for an average of 21.7 days. The average number of days in total spent in hospital across all of a

**Table 1** Average direct medical and transportation costs for managing patients who died from selected NCDs in 2020

| Duration of treatment (days) | Average no of hospital visit (appointments) | Average no of days on admission | Average direct medical costs per person | Average direct non-medical costs (transportation) | Average direct costs per person |
|---|---|---|---|---|---|
| A | B | C | D | E | F=D + E |
| Lung Ca | | | | | |
| 49.2 | 1.6 | 26.7 | ₦484 445.46 | ₦25,707.16 | ₦510 152.62 |
| | | | US$1343.82 | US$71.31 | US$1415.13 |
| Liver Ca* | | | | | |
| 16.6 | 1 | 16.6 | ₦300 764.15 | ₦8186.12 | ₦308,950.27 |
| | | | US$834.30 | US$22.71 | US$857.00 |
| Liver cirrhosis | | | | | |
| 21.7 | 1 | 13.9 | ₦235 131.27 | ₦2990.55 | ₦238 121.83 |
| | | | US$652.24 | US$8.30 | US$660.53 |

*Duration on treatment (management) for liver ca was the same as days on admission for all patients whose records were found because they were all admitted on the day of presentation and stayed on admission until demise.
NCDs, non-communicable diseases.

patient's admissions was 13.9 (1–46) days. Average direct medical cost was ₦235 131.27 (US$652.24) per patient. Average transportation cost was ₦2990.55 (US$8.30). All the patients were residing in the same town as the hospital at the time of presentation. Total direct medical and non-medical cost of managing one patient with liver cirrhosis was estimated to be ₦238 121.83 (US$660.53).

## DISCUSSION

This study on the economic costs of NCDs found that all patients for whom records were found were diagnosed in the terminal stages of disease and the average direct medical costs of care were very high. The mean age of the patients who died from lung cancer was 59.8 (SD=10.6) years. This is comparable with a mean age of 56.6 (SD=21.6) years and range of 30–81 years reported by Ezemba et al in a tertiary hospital in South East Nigeria[30] and 55 years (range: 40–85 years) reported by Adewole et al in a tertiary facility in North Central Nigeria.[31] The mean age of patients with liver cancer in our study was 52.6 (SD=7.9) years, average number of days spent in hospital across all of a patient's admissions was 16.6 days (range: 1–77 days). Patients with liver cirrhosis had a mean age of 44.3 (SD=13.1) years, average days on admission and duration of terminal treatment were 13.9 days and 21.7 days, respectively. These are similar to a mean age of 46.4 (SD=18.0) years reported by Nwokediuko et al in their review of records of patients with liver disease (including liver cancer and liver cirrhosis) admitted between 2005 and 2010 in a tertiary hospital in South East Nigeria.[32] Adekanle et al (2019) similarly documented a mean age of 46.7 (SD=15.3) years, mean duration of admission of 33.4 (SD=26.5) days and a mortality of 47.6% in their retrospective study of patients with liver disease (including hepatocellular cancer and liver cirrhosis, acute hepatitis and metastatic liver disease among others), conducted

between 2013 and 2017 in a teaching hospital in South-west Nigeria.[10]

In our study, all the patients for whom data were available presented in the late stages of disease, hence time between presentation and demise was very short. This would have resulted in a reduction in healthcare costs from hospital admission and transportation for follow-up clinic visits and poorer health outcome. Existing literature has highlighted that access and utilisation of healthcare services is still suboptimal in many low-income and middle-income countries and patients often present late leading to poor disease prognosis.[33–35] Many reasons have been given for delayed presentation of patients in hospital. Patient-related reasons for delayed presentation include inadequate knowledge of the disease and of where to obtain appropriate services, preference for traditional providers over orthodox health facilities, poverty and inadequate transportation.[34 36 37] On the health facility end, experience of the health worker could affect referral for definitive care.[36 37]

The average direct costs (medical and non-medical) for terminal care for each of the diseases studied was very high. We estimated an average direct cost of more than ₦500 000 (₦510 152.62, ie, US$1415.13) for managing a patient with lung cancer. Similarly, other studies have reported high medical costs of care for patients with lung cancer. For example, Edis and Karlıkaya reported that among patients in Turkey, the mean total direct costs per lung cancer patient was more than US$14 000 with costs per patient ranging from US$771.00 to US$104 079.[38] Fan et al reported an average annual direct cost of RMB63 675 ; (US$10 221, US$1=RMB6.23 in 2015) per patient with lung cancer in China. Patients in their study had been diagnosed up to 2 years prior to the study compared with about 49.2 days in our study. Cipriano et al reported that monthly costs in the first 6 months of care for a 72-year-old

diagnosed with lung cancer in the USA in the year 2000, ranged from US$2687 (no active treatment) to US$9360 (chemoradiotherapy) in 2006 prices, although these costs varied by stage at diagnosis and histologic type.[19] Sheenan *et al* in a more recent study reported an estimate of US$21 603 for a month of care in the terminal phase, for a patient who died of lung cancer at age 70 in 2017 in the USA.[18]

In the current study, the average direct cost of managing one patient with liver cancer in Ibadan, Nigeria for approximately 16.6 days in 2020 was estimated to be ₦308 950.27 (US$857.00). Gondek *et al* estimated that the annual cost of HCC in the USA was approximately US$31 641 per patient.[16] The average direct cost of managing one patient with liver cirrhosis for less than a month of terminal care (21.7 days) in our study was estimated to be ₦238 121.83 (US$660.53) and all patients resided in the hospital town. Although these costs are quite substantial, corresponding costs for patients residing out-of-town would be much higher. This is because out-of-town patients would incur additional non-medical costs from higher intercity transportation, short-stay accommodation and upkeep for the patient and accompanying person when they come for hospital appointments. Intangible costs from the stress associated with the intercity travel would also be considerable. In a study of patients with cirrhosis in Vietnam, Nguyen *et al* estimated that the median cost of treatment per session in 2015 was VND7 439 527.40±VND52 514 030.97, that is, approximately US$329.90±US$232.88 (US$1 in VND:US$22 550.18 for 31 December 2015)[39] of which almost 69% of the costs were covered by health insurance.[20]

Although comparisons between our estimates and those provided by the studies cited above need to be made with caution because of our small sample, variations in the disease stage, exact types and duration of treatment provided (among other variables), our findings confirm that irrespective of the setting, the amount expended to treat each of the diseases considered is significant. Boutayeb and Boutayeb in their review of the burden of NCDs in low-income and middle-income countries highlighted the huge difference in costs expended in managing NCDs between patients in developed and low-income and middle-income countries and between affluent and poor patients within developed countries noting that this reflected inequity in access to treatment. In developed countries, a significant proportion of the treatment costs are covered by health insurance and availability of free or largely subsidised care promotes early presentation of patients. This is unlike the situation in Nigeria where the monthly minimum wage for government civil servants was only recently increased to ₦30 000.00—less than US$85.00 in 2019.[40] Treatment costs are also borne out of pocket by the generality of the population in Nigeria and many lower income countries where a significant proportion of the population lives on less than a dollar a day. Thus, these costs are catastrophic for many families and further explain why many patients present late or default after initial presentation and thus their data does not get captured in health facility mortality records.

## CONCLUSION

Our study has demonstrated that direct medical costs of lung and liver cancer and liver cirrhosis in Ibadan, Nigeria are very high, especially when viewed against the prevailing economic environment of the country. Furthermore, many patients in this setting present late resulting in poor prognosis. The substantial costs of care further emphasises the need for greater efforts at prevention of these diseases. Major risk factors for each of the diseases investigated have been identified and interventions exist to reduce engagement in these risky practices. Early diagnosis which can only occur if patients present early and are diagnosed on time needs to be urgently addressed. This would improve disease outcome in the patients and possibly reduce medical costs.

### Limitations

1. The results are based on a relatively small sample, however, they are an important first step to further enquiry.
2. Use of hospital records would have missed other economic costs including costs incurred in other health facilities while trying to obtain a diagnosis, costs of modifications to housing to make homes conducive for patients, indirect and intangible costs. We acknowledge this constraint and limited costs estimates to direct medical and non-medical (transportation) costs only.
3. The hospital data used to compute the average costs of long-term illness were obtained from patients' records in their case notes. However, all the case notes could not be retrieved. This was because patient records are yet to be fully computerised, hence the case notes had to be manually retrieved. However, all case notes were not found because some were yet to be returned to the central records office as they were still being processed administratively. In addition, records found were for patients who had an average survival of less than a month indicating late stage disease. This could have resulted in an under-estimation of the overall costs incurred.
4. Transportation costs were based on the initial addresses documented in the patients' case notes and it was not possible to determine if the patients had relocated closer to the hospital during the course of the illness. This could have led to an overestimation of the costs for the patients who relocated closer to the hospital although costs of relocating to a closer location could offset this difference. The costs, however, still provide a reasonable estimation of the transportation costs.
5. The hospital is a federal government tertiary facility, treatment costs could therefore differ from costs in other federal and private facilities within the country. Information obtained from expert opinion revealed

that, medical costs in the private wing of the study hospital are typically at least three times more than the costs in the main stream hospital and consultation fees and admission fees are considerable higher.

6. Some information such as costs for drug supplements, dietary modifications, costs of seeking for additional medical care from other hospitals, costs of repeat outpatient visits, costs of investigations conducted to arrive at a definitive diagnosis and costs of cancer management received (surgical, chemotherapy and/or radiotherapy) would have been missed.

Inspite of these limitations, our findings are still very useful and represent estimates of costs of terminal care for lung cancer and liver cancer and cirrhosis in our setting.

**Author affiliations**
[1]Institute of Child Health, College of Medicine, University of Ibadan and University College Hospital, Ibadan, Oyo State, Nigeria
[2]Department of Health Management and Economics, School of Public Health, The Hebrew University of Jerusalem, Jerusalem, Israel
[3]Institute of Child Health, College of Medicine, University of Ibadan, Ibadan, Oyo State, Nigeria
[4]Department of Surgery, College of Medicine, University of Ibadan and University College Hospital, Ibadan, Oyo State, Nigeria
[5]Department of Pathology, College of Medicine, University of Ibadan and University College Hospital, Ibadan, Oyo State, Nigeria

**Acknowledgements** All research staff and respondents, management and heads of relevant departments of the University College Hospital, Ibadan are appreciated.

**Contributors** AOO conceptualised and developed the protocol for the main study from which the dataset for the current paper was obtained as this formed the research for her doctoral thesis. AOO conducted the literature search, and was involved in data collection, analysis and write-up of the current paper. AS and OO were AOO's doctoral supervisors. They supervised the entire project and were involved in the write up and revision of the current paper. ESA was involved in data collection and write-up of the data for the current paper. TOA and GOO collated the clinical information of all patients who died in the hospital and generated a list of all-cause mortality for patients managed in the teaching hospital. All authors read and approved the final manuscript.

**Funding** This research was supported by the Consortium for Advanced Research Training in Africa (CARTA). CARTA is jointly led by the African Population and Health Research Center and the University of the Witwatersrand and funded by the Carnegie Corporation of New York (Grant No—G-19-57145), Sida (Grant No:54100113), Uppsala Monitoring Centre and the DELTAS Africa Initiative (Grant No: 107768/Z/15/Z). The DELTAS Africa Initiative is an independent funding scheme of the African Academy of Sciences (AAS)'s Alliance for Accelerating Excellence in Science in Africa (AESA) and supported by the New Partnership for Africa's Development Planning and Coordinating Agency (NEPAD Agency) with funding from the Wellcome Trust (UK) and the UK government.

**Disclaimer** The statements made and views expressed are solely the responsibility of the Fellow.

**Competing interests** None declared.

**Patient consent for publication** Not required.

**Ethics approval** Ethical approval was obtained from the Oyo State Ethics Committee (AD13/479/193). Permission to retrieve and review patients' mortality records and case notes was obtained from the University College Hospital management and the heads of the relevant departments. The department heads were assured that information which could identify any patient would not be taken.

**Provenance and peer review** Not commissioned; externally peer reviewed.

**Data availability statement** No data are available. The datasets generated and/or analysed during the current study are not publicly available but are available from the first author on reasonable request.

**ORCID iD**
Adesola Oluwafunmilola Olumide http://orcid.org/0000-0003-4372-9822

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
