## [Reviewer comments · BMJ Open]

ARTICLE DETAILS

TITLE (PROVISIONAL)	Economic costs of terminal care for selected non-communicable diseases from a healthcare perspective: A retrospective review of mortality records from a tertiary hospital in Nigeria
AUTHORS	Olumide, Adesola; Shmueli, Amir; Omotade, Olayemi; Adebayo, Emmanuel; Alonge, Temitope; Ogun, Gabriel

VERSION 1 – REVIEW

REVIEWER	Sanjay K Mohanty International Institute for Population Sciences, Mumbai, India
REVIEW RETURNED	21-Oct-2020

GENERAL COMMENTS	The paper lack sufficient number of cases for generalisation and drawing inferences. The paper can be at best a note and can be resented qualitatively. It should be lined with demographic and socio-economic characteristic of patients from case record. table 1 is not required. Othe tables can be prenetd in terms of graph Introduction, motivation is excellent but the paper suffers from analyses and minimum number. It will not be justice to readers to recommend with such low sample size
--

REVIEWER	David Goldsbury Cancer Council NSW, Australia
REVIEW RETURNED	14-Jan-2021

GENERAL COMMENTS	Thank you for the opportunity to review this manuscript. The study describes the economic costs of terminal care for patients in a Nigerian hospital who die from lung cancer or liver diseases. The paper is generally well written and covers a very important and interesting topic, and with some clarifications it will make for a very useful resource. An initial concern is the small sample involved. There were only 5 lung cancer patients, 9 liver cancer patients, and 8 liver cirrhosis patients. While they provide valuable information and a baseline for future work, it might be misleading to suggest these are generalisable to all people with these conditions. The sample size should be noted in the abstract. That being said, given the absence of any other literature in the area, this sample of 22 is a big step forward. Secondly, all conditions have an average of less than one month of follow-up. It might not be appropriate to extrapolate this to an average monthly cost, as these might include costs that are only relevant to the final days/weeks of life and would not have been incurred if patients had presented earlier. Perhaps just use costs per
--

	patient and note that the average time under study was less than one month. This is still fine for the comparisons that are included in the discussion. The abstract describes the minimum wage and this provides excellent context for the cost results. This should also be included in the main text. It would also be useful for context to have the average numbers of cases and/or deaths for each condition in the region/country, and any information on expected survival times and the representativeness of the sample. The description of time in hospital needs to be clarified, in particular the meaning of “on admission” used throughout the manuscript. For example, does “average days on admission” mean the average number of days in total spent in hospital across all of a patient’s admissions, or is it average number of days per admission? The results say lung cancer patients were “managed for 0.92 months (from diagnosis to demise) and on admission for 27.4 days” – this seems to be the same amount of time expressed in different units. Are they the same (i.e. patients were in hospital the entire time from diagnosis to death) or is there a difference that needs to be explained? For liver cancer the average was “0.54 months (approximately 21 days)”, wouldn’t this be $0.54 \times 365 / 12 = 16$ days? Also, 0.54 months is mentioned three times in three sentences, should re-write to avoid repetition. For liver cirrhosis, all patients lived in the hospital town. What is the likely scenario and related costs for people from other areas? Greater transport costs and maybe fewer/more hospital days? Some other noted limitations could be expanded upon. Can the authors offer any thoughts on potential public/private cost differences for all conditions, and the types of records that might have been missed for the included cases? The discussion section should start with some of the highlights or key findings The other notes are all minor. The authors say that cost of illness studies always estimate the “maximum amount” that could be saved – this is not always true, but they can help with the estimation of an amount, suggest leaving out “maximum”. Spell out “non-communicable diseases” in the first use of NCD in the abstract. The introduction describes the economic burden “in our clime” – perhaps this should be “in our region”. The results describe a “male:female ration”, the latter word should be “ratio”. Perhaps more importantly, the ratios are actually giving the actual numbers of cases (e.g. 3:2 is $n=3$ and $n=2$) and it would be better for the sake of clarity to just list the numbers. Table 1 is described sufficiently in the text, so only one of these is required (either text or table).
--	--

VERSION 1 – AUTHOR RESPONSE

Reviewer 1	
The paper lack sufficient number of cases for generalisation and drawing inferences. The paper can be at best a note and can be resented qualitatively.	Thank you very much for this comment. The authors agree that the numbers are small and we therefore do not attempt to generalize. We have highlighted the small sample representing about 24% of the total mortality during the period in the limitations box and also in the limitations section in the main body of the write-up. We have also stated that the costs are not generalizable but provide reasonable estimates of costs in the study site. We however believe the findings are an important first step to further enquiry, especially as published information is really sparse. We have also presented only descriptive data and not conducted any inferential statistical tests.
It should be lined with demographic and socio-economic characteristic of patients from case record.	Thank you very much for the comment. We presented demographic information on the patients that we felt were relevant for estimating the economic costs. We have revised the data Their current socio-economic situation would not have been reflected in the case notes because they were all in the terminal stages and would have stopped working irrespective of the occupation they mentioned at presentation.
Table 1 is not required.	Table 1 has been deleted
Other tables can be presented in terms of graph	We have tried to convert the tables to graphs but find that a lot of information is lost, we would like to thus make a case to retain the tables. The tables have been merged into one composite table
Introduction, motivation is excellent but the paper suffers from analyses and minimum number. It will not be justice to readers to recommend with such low sample size	We appreciate this comment. We have included the small sample size as a limitation
Reviewer 3	
An initial concern is the small sample involved. There were only 5 lung cancer patients, 9 liver cancer patients, and 8 liver cirrhosis patients. While they provide valuable information and a baseline for future work, it might be misleading to suggest these are generalisable to all people with these conditions. The sample size should be noted in the abstract. That being said, given the absence of any other literature in the area, this sample of 22 is a big step	Thank you for this comment. We have included the sample size of 22 in the abstract.

forward.	
Secondly, all conditions have an average of less than one month of follow-up. It might not be appropriate to extrapolate this to an average monthly cost, as these might include costs that are only relevant to the final days/weeks of life and would not have been incurred if patients had presented earlier. Perhaps just use costs per patient and note that the average time under study was less than one month. This is still fine for the comparisons that are included in the discussion.	We have corrected this in line with the reviewer comments and used costs per patient and specified the average duration of time on treatment documented in the case notes (from presentation in hospital to demise) in the abstract and results.
The abstract describes the minimum wage and this provides excellent context for the cost results. This should also be included in the main text.	Thank you for this comment. We have included the information in the discussion “This is unlike the situation in Nigeria where the monthly minimum wage for government civil servants is NGN 30,000 - less than USD 85”
It would also be useful for context to have the average numbers of cases and/or deaths for each condition in the region/country, and any information on expected survival times and the representativeness of the sample.	Thank you for this comment. We have provided additional information in the Introduction
The description of time in hospital needs to be clarified, in particular the meaning of “on admission” used throughout the manuscript. For example, does “average days on admission” mean the average number of days in total spent in hospital across all of a patient’s admissions, or is it average number of days per admission?	On admission refers to, “the average number of days in total spent in hospital across all of a patient’s admissions” We have clarified this in the tables
The results say lung cancer patients were “managed for 0.92 months (from diagnosis to demise) and on admission for 27.4 days” – this seems to be the same amount of time expressed in different units. Are they the same (i.e. patients were in hospital the entire time from diagnosis to death) or is there a difference that needs to be explained?	This has been corrected. It has also been changed to, “from presentation with symptoms to demise”. The duration of management = 49.2 days; admission = 26.7 days.
liver cancer the average was “0.54 months (approximately 21 days)”, wouldn’t this be $0.54 \times 365 / 12 = 16$ days? Also, 0.54 months is mentioned three times in three sentences, should re-write to avoid	Liver Ca. This has been revised to 16.6 days. All the patients whose case notes could be retrieved, were patients admitted and who were on admission from then until demise. Hence the dates between presentation and demise and the days on admission were the same. This

repetition.	information has been included in the write-up. We have also changed the 0.54 months to 16.6 days.
For liver cirrhosis, all patients lived in the hospital town. What is the likely scenario and related costs for people from other areas? Greater transport costs and maybe fewer/more hospital days?	Additional information has been provided in the discussion as presented below. The average direct cost of managing one patient with liver cirrhosis for less than a month of terminal care in our study was estimated to be NGN 238,121.83 (USD 660.53) and all patients resided in hospital town. Although these costs are quite substantial, corresponding costs for patients residing out of town would be much higher. This is because out-of-town patients would incur additional non-medical costs from higher inter-city transportation, short-stay accommodation and upkeep when they come for hospital appointments. Intangible costs from the stress associated with the inter-city travel would also be considerable.
Some other noted limitations could be expanded upon. Can the authors offer any thoughts on potential public/private cost differences for all conditions, and the types of records that might have been missed for the included cases?	We have expanded the discussion in the limitations section Limitation 1: Use of hospital records would have missed other economic costs including costs incurred in other health facilities while trying to obtain a diagnosis, costs of modifications to housing to make homes conducive for patients, indirect and intangible costs. We acknowledge this constraint and limited costs estimates to direct medical and non-medical (transportation) costs only Cost of care in a private facility would have been much higher than public costs. We have included the following statement in limitation 4, "Medical costs in the private wing of the study hospital are typically at least three times more than the costs in the main stream hospital and consultation fees and admission fees are considerable higher". The types of records that might have been missed for the included cases have been mentioned in limitation 5: "Some information such as costs for drug supplements, dietary modifications, costs of seeking for additional medical care from other hospitals, costs of repeat out-patient visits, costs of investigations conducted to arrive at a definitive diagnosis and costs of cancer management received (surgical, chemotherapy and/ or radiotherapy) would have been missed.
The discussion section should start with some of the highlights or key findings	Thank you for this comment, we have commenced the discussion with key findings.
The other notes are all minor. The authors say that cost of illness studies always estimate the "maximum amount" that could be saved – this is not always true, but they can help with the estimation of an amount, suggest leaving out "maximum	Thank you very much for this suggestion. We have deleted the word "maximum".
Spell out "non-communicable diseases" in the first use of NCD in	We have made this correction

the abstract.	
The introduction describes the economic burden “in our clime” – perhaps this should be “in our region”.	“in our clime” – changed to “in our region”.
The results describe a “male: female ration”, the latter word should be “ratio”. Perhaps more importantly, the ratios are actually giving the actual numbers of cases (e.g. 3:2 is n=3 and n=2) and it would be better for the sake of clarity to just list the numbers.	This has been corrected and the values updated
Table 1 is described sufficiently in the text, so only one of these is required (either text or table).	Table 1 has been deleted
For Appendix Table 1, what is CTTD?	Full meaning – Closed Tube Thoracostomy Drainage. This has been included as a footnote in the table

VERSION 2 – REVIEW

REVIEWER	David Goldsbury Cancer Council NSW, Australia
REVIEW RETURNED	16-Mar-2021

GENERAL COMMENTS	Thank you to the authors for the revisions to the manuscript, they have done a very good job and I think only some minor adjustments are needed from here, as described below. The manuscript should note, as described in the authors’ responses to the reviewers, that these results are an important first step to further enquiry. The results are very useful but they are based on a relatively small sample (as has been explained in the manuscript). The introduction says that liver and lung cancer have incident rates in Nigeria of 4.4% and 1.4% respectively. Please clarify if this is lifetime risk in the entire population, or risk over some other time period, or the proportion that they comprise of new cancer cases in Nigeria, or something else. The introduction also lists results mean annual costs from Muka et al for cancer and CVD – please clarify if these are costs per case. The methods note that within the study sample the “average [follow-up time] ... ranged from 0 to about four and a half months”. Using average and range together is confusing here. Should the word “average” be dropped? Or does the range refer to the different averages (means?) for the three different conditions, or is it the ranges, or inter-quartile ranges, or some other measure for all? The results and discussion describe the cohort mean ages as X +/- Y years – please clarify the measure for the latter figure, is it the standard deviation? And the first sentence of the results should note that there were eight patients with liver cirrhosis – it is included elsewhere but this one word would simplify reading and save looking back through the manuscript.
---

	The discussion compares the results to “very high direct medical costs” from Enstone et al – can some numerical value be given here? Limitation number 4 describes the higher private costs as “typically at least three times more”, is there a reference for this or is it expert opinion? The footnote in Table 1 should be about liver cancer, not lung cancer.
--	--

VERSION 2 – AUTHOR RESPONSE

Reviewer 1	
Thank you to the authors for the revisions to the manuscript, they have done a very good job and I think only some minor adjustments are needed from here, as described below.	We appreciate all comments given. These have contributed towards the improved version.
The manuscript should note, as described in the authors’ responses to the reviewers, that these results are an important first step to further enquiry. The results are very useful but they are based on a relatively small sample (as has been explained in the manuscript).	We have included this information in the Article summary (below the abstract) Strengths and limitations highlights, bullet point five:  - In spite of the small sample, the findings are reasonable estimates of direct costs of terminal care for the selected NCDs in the study setting and are an important first step to further enquiry. And as limitation after the conclusion Limitations:  1. The results are based on a relatively small sample, however, they are an important first step to further enquiry.
The introduction says that liver and lung cancer have incident rates in Nigeria of 4.4% and 1.4% respectively. Please clarify if this is lifetime risk in the entire population, or risk over some other time period, or the proportion that they comprise of new cancer cases in Nigeria, or something else.	We have revised the statement and included the statement below in brackets as was stated in the source document - the Global cancer observatory (Weighted/simple average of the most recent local rates applied to 2020 population). “Liver and lung cancer were among the top 20 causes of cancer in Nigeria with incidence rates of 4.4% and 1.4% respectively (based on weighted/simple average of the

	most recent local rates applied to 2020 population) (9)
The introduction also lists results mean annual costs from Muka et al for cancer and CVD – please clarify if these are costs per case.	Thank you very much for this comment. We have revised the statement to include information that these are costs per patient. Muka et al (2015) in their review of the global impact of non-communicable diseases on healthcare found that globally, cancer and cardiovascular diseases (CVD) had the highest reported mean annual total direct costs per patient of approximately 197,772 USD and 81,096 USD, respectively.
The methods note that within the study sample the “average [follow-up time] ... ranged from 0 to about four and a half months”. Using average and range together is confusing here. Should the word “average” be dropped? Or does the range refer to the different averages (means?) for the three different conditions, or is it the ranges, or inter-quartile ranges, or some other measure for all?	The word “average” has been dropped.
The results and discussion describe the cohort mean ages as X +/- Y years – please clarify the measure for the latter figure, is it the standard deviation? And the first sentence of the results should note that there were eight patients with liver cirrhosis – it is included elsewhere but this one word would simplify reading and save looking back through the manuscript.	The latter figure in all cases is standard deviation. “Standard deviation” has been included for clarity. We have included the statement, “there were eight patients with liver cirrhosis”, in the third sentence under results. This comes just before the information on the mean age and standard deviation for patients with liver cirrhosis. The statement has been revised to read, The mean age (and standard deviation) of the five patients (three male and two female) with a diagnosis of lung cancer was, 59.8 (±10.6) years. The patients (eight males and one female) with liver cancer had an average age of 52.6 years (standard deviation = 7.9). There were eight male patients with liver cirrhosis and

	they had a mean age of 44.3 years (standard deviation = 13.1).
The discussion compares the results to “very high direct medical costs” from Enstone et al – can some numerical value be given here?	We revised this slightly. The systematic review by Enstone et al references several papers. We selected one of these and revised the statement to read. For example, Edis and Karlikaya (2007) reported that among patients in Turkey, the mean total direct costs per lung cancer patient was more than USD 14,000 with costs per patient ranging from USD 771.00 to USD 104,079 (38). Other literature that had been cited remained the same
Limitation number 4 describes the higher private costs as “typically at least three times more”, is there a reference for this or is it expert opinion?	We appreciate this comment. The statement has been revised to include the suggestion provided: Information obtained from expert opinion revealed that, medical costs in the private wing of the study hospital are typically at least three times more than the costs in the main stream hospital and consultation fees and admission fees are considerable higher.
The footnote in Table 1 should be about liver cancer, not lung cancer.	This was a mistake and has been changed to liver cancer.